# Preparation of Calcium Phosphate Compounds on Zirconia Surfaces for Dental Implant Applications

**DOI:** 10.3390/ijms23126675

**Published:** 2022-06-15

**Authors:** Mei-Shuan Cheng, Eisner Salamanca, Jerry Chin-Yi Lin, Yu-Hwa Pan, Yi-Fan Wu, Nai-Chia Teng, Ikki Watanabe, Ying-Sui Sun, Wei-Jen Chang

**Affiliations:** 1School of Dentistry, College of Oral Medicine, Taipei Medical University, Taipei 110, Taiwan; m204109008@tmu.edu.tw (M.-S.C.); eisnergab@hotmail.com (E.S.); drjerrylin@gmail.com (J.C.-Y.L.); shalom.dc@msa.hinet.net (Y.-H.P.); yfwu@tmu.edu.tw (Y.-F.W.); dianaten@tmu.edu.tw (N.-C.T.); 2Department of Oral Medicine, Infection and Immunity, Harvard School of Dental Medicine, Boston, MA 02115, USA; 3Department of General Dentistry, Chang Gung Memorial Hospital, Taipei 110, Taiwan; 4Graduate Institute of Dental & Craniofacial Science, Chang Gung University, Taoyuan 333, Taiwan; 5School of Dentistry, College of Medicine, China Medical University, Taichung 404, Taiwan; 6Dental Department, Taipei Medical University Hospital, Taipei 110, Taiwan; 7Department of Gerodontology, Tokyo Medical and Dental University, Tokyo 113-8510, Japan; ikki.ore@tmd.ac.jp; 8School of Dental Technology, College of Oral Medicine, Taipei Medical University, Taipei 110, Taiwan; 9Dental Department, Taipei Medical University, Shuang-Ho Hospital, Taipei 110, Taiwan

**Keywords:** zirconia, surface modification, alkali treatment, calcium phosphate coating, sintering

## Abstract

Titanium is widely used in medical implants despite the release of heavy metal ions over long-term use. Zirconia is very close to the color of teeth; however, its biological inertness hinders bonding with bone tissue. Alkaline treatment and coatings of calcium phosphate can be used to enhance bone regeneration adjacent to dental implants. This study examined the effects of alkaline treatment, calcium phosphate coatings, and sintering, on the physical properties of implant material. Our analysis confirmed that the calcium phosphate species were octacalcium phosphate (OCP). The sintering of calcium phosphate was shown to create B-type HAP, which is highly conducive toward the differentiation of mesenchymal stem cells (MSCs) into osteoblasts for the facilitation of bone integration. Conclusions: This study demonstrated the room-temperature fabrication of dental implants with superhydrophilic surfaces to enhance biocompatibility.

## 1. Introduction

### 1.1. Titanium

In the last few years, titanium has become the material of choice for dental implants, due to its excellent biocompatibility; however, it is prone to oxidative corrosion when it comes into contact with fluoride or other metal alloys within an acidic oral environment. The gray color of titanium implants can also penetrate the tissues surrounding the implant, forming a dark shadow, particularly when placed in an area with thin gums. This has prompted interest in the use of zirconia for long-term dental applications [1].

### 1.2. Zirconia

Zirconia is aesthetically pleasing, non-toxic, corrosion resistant, and highly biocompatible. The constraint of the tetragonal-to-monoclinic phase transformation leads to the high fracture toughness that is exhibited by zirconia. However, the biological inertness of zirconia surfaces is not conducive to the formation of chemical bonds with natural bone. Surface treatment is required to overcome these issues [2].

### 1.3. Surface Treatment

#### 1.3.1. Alkaline Treatment

Researchers have developed a variety of surface modification techniques that are aimed at improving bone integration and reducing healing time. It has been established that surface modification can promote cell adhesion, growth, osteogenesis, and cell differentiation [3]. Alkaline treatments are a low-cost approach for achieving a uniform surface profile while providing surface OH^−^ functionalization to enable the precipitation of Ca^2+^ from body fluids. It also facilitates the binding of amino groups (NH_2_) in bone, thereby inducing the precipitation of hydroxyl apatite for osseointegration [4,5,6].

#### 1.3.2. Calcium Phosphate Coatings

One approach to overcoming the biological inertness of zirconia involves coating the material with calcium phosphate. Calcium phosphate is widely used in bone tissue engineering, due to its bone conductivity and the fact that it possesses many of the properties of natural bone. It can also induce the differentiation of mesenchymal stem cells (MSCs) into osteoblasts and osteoclasts to facilitate osseointegration [7]. Unfortunately, calcium phosphate coatings lack stability and bonding strength with the underlying substrate. Various physicochemical methods have been developed for the deposition of calcium phosphate coatings, including plasma spraying, electrophoretic deposition, pulsed laser deposition, and immersion. Plasma spraying is a rapid, low-cost, high-temperature process that is widely used to deposit hydroxyapatite onto orthopedic implants. Unfortunately, hydroxyapatite coatings exceeding 50 μm are prone to delamination following thermal decomposition. Soaking is a low-temperature method that is applicable to substrates of any shape while allowing for the formation of bands with growth factors, antibiotics, etc. [8].

#### 1.3.3. Sintering Treatment

Researchers have determined that sintering can increase the compactness of zirconia, thereby making it more conducive to bonding with bone tissue [9]. Note that sintering involves pressing a solid material to form a shape under the effects of moderate heat (below the melting point of the material). Sintering reduces porosity while increasing strength, electrical conductivity, thermal conductivity, and transparency. The energy source in sintering is the surface free energy or the chemical energy [10].

##### Sintering Zirconia

The sintering of zirconia can affect the optical properties of the substrate, including the clarity, contrast, color, and light transmittance. Increasing the sintering temperature to 1300 °C while reducing the sintering time can reduce the grain size and increase transparency without altering the hardness or the bending strength. Increasing the sintering temperature further increases the grain size while reducing the bending strength. Lengthening the sintering time decreases transparency, resulting in a darker reddish hue, indicating its transformation into a single oblique crystal with high surface roughness [11]. Higher sintering temperatures and extended sintering times, respectively, lead to an increase in the size of zirconia grains, a decrease in the yttrium content, and/or an increase in the content of a single oblique phase, thereby increasing susceptibility to aging [12].

##### Sintering Calcium Phosphate

The solubility of calcium phosphate varies as a function of the calcium/phosphorus ratio [13]. The temperature and pressure associated with sintering can alter the grain size and shape of the material, with corresponding effects on the material properties. The fact that the melting point of calcium phosphate is 1391 °C means that pressure-free sintering should be performed at temperatures ranging from 1000 to 1350 °C [14].

The composition of calcium phosphate is similar to that of bone and teeth; however, it tends to be somewhat brittle. Thus, our objective in this study was to create a composite material by which to combine the aesthetic appeal and fracture toughness of zirconia with the biological activity of calcium phosphate, to promote osseointegration.

## 2. Results

Figure 1 is the graphic abstract of this study. The outline of the results of this experimental design can be seen in the graphic abstract.

### 2.1. Optimal Sodium Hydroxide Concentration

The SEM images in Figure 2 revealed that the deep and wide grooves of the negligible cavity in specimens A4, A5, and A6 are well-suited to bonding with calcium phosphate. This initial analysis indicated that the optimal NaOH concentration would be 2.5 M. The EDS results in Figure 2 revealed that the concentration of NaOH was inversely proportional to the Zr content. XPS analysis confirmed the EDS results.

We determined that the depths of the holes in A1, A2, and A3 were less than 1.7 nm (Figure 3), whereas the depths of the holes in A4, A5, and A6 exceeded 49 nm. Note that the hole depth was correlated with the strength of the calcium–phosphate bonds. We accordingly selected for subsequent analysis specimens that had been treated with higher concentrations of NaOH (2.5 M, 5.0 M, and 10 M). FTIR analysis (Figure 4) revealed that NaOH concentrations were correlated with the strength of the OH^−^ signal and the corresponding adsorption of Ca^2+^ ions on the surface. Contact angle analysis revealed that specimens A1 and A6 were superhydrophilic, with no significant difference among them (Figure 2). Thus, the groups selected for subsequent analysis were A4, A5, and A6.

Roughness analysis using white light interferometry (Figure 2) revealed that specimens A4, A5, and A6 were rougher than specimens A1 and A2. Analysis using the student’s t-test statistic (*p* < 0.05) revealed that the roughness of specimen A6 was significantly higher than that of A5. No significant differences were observed between specimens A4 and A5 in terms of roughness. Note that surface roughness is beneficial to bonding with calcium phosphate. Taken together, these results indicate that the optimal treatment conditions were NaOH (2.5 M), followed by Na_3_PO_4_ (5.0 M) [15].

In Figure 5, we observed that a 2θ of 30, 34, 35, 50, and 51 degrees indicated the presence of tetragonal crystals of zirconia, whereas there was no monoclinic crystal phase.

Based on the above results, we selected specimens A4 and A5 for subsequent experiments involving Ca(OH)_2_.

### 2.2. Optimal Concentration of Calcium Hydroxide

The SEM diagram in Figure 6 revealed that the quantity of Ca deposited on the surface of specimen A4 exceeded that of specimen A5. Note that the availability of surface calcium deposits is crucial to the adhesion of negatively charged PO_4_^3^^−^ ions. We therefore selected groups A4-1, A4-2, and A4-3 for subsequent experiments involving the immersion of specimens in Na_3_PO_4_.

The EDS analysis in Figure 6 revealed more Ca signals in groups A4-2 and A4-3. XPS analysis revealed a positive correlation between the strength of the Ca_2p_ signal and the concentration of Ca(OH)_2_ in the immersion, thereby supporting our identification of groups A4-2 and A4-3 as being optimal. XPS analysis also revealed the highest calcium content in specimen A4-3. These results confirm our selection of groups A4-2 and A4-3 for subsequent experiments (Figure 7).

FTIR analysis in Figure 8 revealed that the strength of the surface OH^−^ signal was correlated with the concentration of Ca(OH)_2_. Note that the OH^−^ groups attract calcium ions in natural bones to facilitate integration with the implants. This provided further support for our selection of specimens A4-2 and A4-3. The statistical roughness chart in Figure 6 revealed an inverse correlation between the concentration of Ca(OH)_2_ and the roughness. The fact that the roughness values of specimens A4-1, A4-2, and A4-3 were all relatively high gave us no reason to disqualify A4-2 or A4-3 as being optimal. All six groups presented superhydrophilicity without significant differences among them (Figure 6). In Figure 9, we observed that a 2θ of 30, 34, 35, 50, and 51 degrees indicates the presence of tetragonal crystals of zirconia, whereas there is no monoclinic crystal phase (Figure 8). Based on the above results, we selected A4-2 and A4-3 as the best conditions.

### 2.3. Optimal Concentration of Sodium Phosphate

The SEM diagram in Figure 10 revealed the largest pores in groups A4-2-1 and A4-3-1, which could presumably provide access for the transport of nutrients and bone cells, to promote osseointegration. The EDS results in Figure 10 were used to calculate the calcium/phosphorus ratio in each group to derive the surface elements of the coatings.

Fourier-transform infrared spectroscopy results (Figure 11a–d) revealed that the strength of the PO_4_^3^^−^ signals was correlated with the concentration of NaPO_4_. Contact angle analysis revealed that treatment with sodium phosphate did not alter the superhydrophilic properties, regardless of the concentration.

In our crystallographic analysis of specimens A4-2-1 and A4-3-1, we observed a peak characteristic of calcium phosphate at a 2θ angle of 27 degrees (Figure 11c,d). We also observed a faint signal at 53 degrees. Taken together, we can infer that the surface sediment was calcium phosphate. The specimens with the highest roughness values were A4-2-1 and A4-3-1, which were subsequently selected for further analysis.

### 2.4. Analysis of Calcium Phosphate on the Surface of Specimens

According to the literature, pure CaP coatings exhibit poor stability and weak binding strength to the substrate [16]. In the current study, we sought to overcome these shortcomings via the application of CaP on zirconia specimens that had previously undergone alkali treatment to etch surface features by which the CaP could adhere securely. Note that infrared spectrograms of calcium phosphate synthesized in previous studies contained polar molecules on the surface, thereby imbuing the materials with superhydrophilic properties.

Calcium phosphate powder is difficult to dissolve in water in uniform concentrations [8]. Researchers have also demonstrated that if solutions of Ca(OH)_2_ and Na_3_PO_4_ are prepared separately, it is difficult to control the rate of precipitation, thereby making it difficult to quantify the amount of calcium phosphate on the specimens [17]. In the current study, we sought to overcome this issue by adopting the chemical precipitation method, which involved soaking specimens in NaOH to produce surface OH^-^ groups by which to attract calcium ions in the form of calcium phosphate.

As shown in the SEM diagrams, the calcium phosphate particles were evenly distributed on the surface without interfering with the porosity of the underlying substrate.

The coating on specimen A4-3-1 was significantly thicker than that on specimen A4-2-1; however, the pitting was not as extensive. The Ca/P ratio obtained from specimen A4-2-1 was 1.33 (indictive of OCP), whereas the Ca/P ratio of A4-3-1 was 1.6 (indicative of HAP) (Figure 10). The X-ray spectra from A4-3-1 did not present signals indicative of Zr, presumably due to the thickness of the surface coating. Note that these peaks were comparable to those of JCPDS file NO. 9-432; i.e., the standard peak of octacalcium phosphate (Figure 10) [18]. Fourier transform infrared energy spectrograms in Figure 10 revealed OH^−^ groups at wavelengths of 3360 cm^−1^ and 1650 cm^−1^. We also detected a signal at a wavelength of 1020 cm^−1^, which is indicative of PO_4_^3^^−^. Note that calcium phosphate has a characteristic peak at a wavelength of 1020 cm^−1^. The absorption peak obtained from specimen A4-2-1 was similar to that of octacalcium phosphate. The absorption peaks observed in A4-3-1 were identical to those of the octacalcium phosphate and apatite (ns-HAP) occurring in natural bone [10]. Overall, the roughness of the calcium phosphate coatings synthesized in this study was in the range of 1.5 μm, which is well-suited to protein adhesion (Figure 10) [19].

The XRD results in Figure 11revealed peaks that were characteristic of HAP at approximately 26, 28, 29, 30–35, 39, 46, 49 and 50° (2θ). Overall, our results indicated that the surface coating on specimen A4-2-1 was octacalcium phosphate, whereas the coating on specimen A4-3-1 was HAP.

Note that long-term immersion is a strong base. We therefore conducted experiments that were aimed at identifying the optimal operating conditions in terms of alkali concentration, immersion temperature, and immersion duration.

### 2.5. Surface Properties of Various Materials after Sintering

The SEM diagram in Figure 11 revealed pits on the surface of the zirconia (Z) sintered at 1000 or 1100 °C (Z L and Z H). Nonetheless, the same samples presented a reduction in the number of pores (i.e., densification), compared to the unsintered samples. The calcium phosphate was more firmly attached to specimen A4-2-1 than to specimen A4-3-1, presumably due to dehydrogenation during sintering.

We observed no significant difference in the compositions of the coatings on specimen A4-2-1 before and after sintering. Note, however, that the Ca/P ratio of A4-3-1 increased with an increase in the sintering temperature (Figure 12). After sintering, the peak, indicative of an amide I bond at 1637 cm^−1^, nearly disappeared, and peaks appeared at 1023 cm^−1^ and 1092 cm^−1^, indicating the stretching vibration band of the phosphate group (PO_4_^3-^). The peaks appearing at 1460 cm^−1^ and 1520 cm^−1^ can be attributed to asymmetric low-wave CO_3_^2^^−^ (Figure 13). According to the literature, if CO_3_^2^^−^ is located at 1548 cm^−1^, then the apatite is type A. If CO_3_^2^^−^ are located at 1410 cm^−1^ and 1452 cm^−1^, then the apatite is type B. If CO_3_^2^^−^ are located at 1456 cm^−1^ and 1541 cm^−1^, then the apatite is type AB [14]. Our results revealed that after sintering, the surface coatings of specimens A4-2-1 and A4-3-1 were B-type HAP. The stretching band of water molecules gradually decreased after sintering, which indicates that the calcium phosphate underwent dehydration during the sintering process, resulting in the formation of β-TCP. The process of dehydration can be described using the following reaction equation [9]:Ca_10_(PO_4_)_6_(OH)_2_ → Ca(PO_4_)_6_(OH)_2−2x_ + xH_2_O(1)

Before and after sintering, there was no significant difference in the hydrophilicity of the surface of the material, and the superhydrophilicity was still maintained (Figure 14).

## 3. Discussion

### 3.1. Implant Materials

Dental implants are currently the best solution for dealing with tooth loss; however, they is prone to the release of metal ions, and lack aesthetic appeal [2]. The color of zirconia implants is close to that of natural teeth. They also provide excellent chemical stability, as well as high fracture resistance, bending strength, and corrosion resistance. Zirconia also stimulates the formation of osteoblasts and osseointegration, with an osseointegration index that is comparable to that of titanium implants [1].

### 3.2. Alkaline Treatment

Scholars have developed numerous surface modification techniques for zirconia [3], including sandblasting, acid etching, and alkaline treatment. The effects of sandblasting depend largely on the pressure, particle size, type, and blasting distance. The effects of acid or alkali etching depend on the concentration and reaction time [20,21]. The engravings produced via sandblasting tend to be uneven and irregular, whereas those produced via etching tend to be uniform and regular. Sandblasting produces indentations of greater depth; however, this approach also tends to introduce a high concentration of residual metal ions. NaOH etching at high concentrations can be used to create features up to 6 μm in depth; however, the features also tend to be very wide (up to 80 μm). Sandblasting tends to reduce the hydrophilicity of the material, whereas etching is generally able to preserve hydrophilicity [20]. In the current study, our use of three solutions facilitated the adhesion of polar molecules on the surface, thereby preserving the superhydrophilic properties to facilitate bone cell attachment. Sandblasting also tends to change the zirconia from a square crystal phase to a monoclinic crystal phase, thereby causing the material to age, with a corresponding decrease in mechanical properties [2].

### 3.3. Calcium Phosphate Coatings

The fact that calcium phosphate is the main inorganic component of bone tissue makes it highly biocompatible and well-suited to osseointegration. Calcium phosphate coatings also provide good corrosion resistance and antimicrobial properties [13,16,22]. The calcium phosphate species synthesized in the current study were OCP and HAP. OCP has a triclinic crystal structure. It is present in human teeth and is believed to form during the initial stages of HAP formation in bone mineralization. HAP (i.e., Ca_10_(PO_4_)_6_(OH)_2_) is abundant in human bones, and a Ca/P ratio of 1.67 makes it the most stable form of calcium phosphate. HAP does not cause inflammation in the surrounding tissues.

### 3.4. Sintering Treatment

Sintering is meant to increase the density of materials to make it easier for bone cells to attach. The sintering process in the current study resulted in the formation of calcium carbonate phosphate matching the composition of natural bone [14]. The superhydrophilic surface can assist in angiogenesis in the early stages of osteogenesis and facilitate osseointegration. The B-type HAP synthesized in the current study bonds easily to bone cells. Sintering was also shown to eliminate the formation of monoclinic zirconia, CaO, and α-Ca_3_(PO_4_)_2_, all of which can have a detrimental effect on its mechanical properties [16].

## 4. Materials and Methods

### 4.1. Specimen Preparation

Discs of zirconia (diameter = 25 mm, thickness = 2 mm) (EPILEDS (Kuang Hung, Tainan) underwent ultrasonic cleaning in purified water (DC200H; DELTA, Taiwan) for 15 min, followed by oven drying for 24 h. This group is indicated by the letter Z.

### 4.2. Surface Treatment

#### 4.2.1. Alkaline Treatment

##### Alkaline Treatment in Sodium Hydroxide (NaOH)

In accordance with the results in previous studies, we prepared NaOH solutions in the following concentrations: 0.25 M (A1), 0.5 M (A2), 1.0 M (A3), 2.5 M (A4), 5.0 M (A5), and 10.0 M (A6) [21]. The zirconia disc was immersed in the various sodium hydroxide concentrations in a 6-well plate at room temperature for 24 h. The specimens then underwent three cycles of ultrasonic cleaning using purified water for 30 min, followed by oven drying for 24 h.

The formula for the NaOH chemical reaction during immersion was as follows:ZrO_2_ + 2NaOH → 2NaO + Zr(OH)_2_(2)
2NaOH + CO_2_ → Na_2_CO_3_ + H_2_O(3)

#### 4.2.2. Coating Calcium Phosphate

##### Stage 1: Soaking in Calcium Hydroxide Solution (Ca(OH)_2_)

This experiment was performed using specimens that were previously soaked in NaOH at concentrations of 2.5 M (A4) or 5.0 M (A5). The specimens were immersed for 24 h in Ca(OH)_2_ using various concentrations of Ca(OH)_2_: 1.0 M (A4-1 and A5-1), 2.5 M (A4-2 and A5-2), and 5.0 M (A4-3 and A5-3). Note that the 10.0 M solution was not used, due to its excessive viscosity. The specimens then underwent three cycles of ultrasonic cleaning in purified water for 30 min, followed by oven drying for 24 h.

The formula for the Ca(OH)_2_ chemical reaction during immersion was as follows:Ca(OH)_2_ + Na_2_CO_3_ → CaCO_3_↓ + 2NaOH(4)
Ca(OH)_2_ + CO_2_ → CaCO_3_↓ + H_2_O(5)

##### Stage 2: Soaking in Sodium Phosphate Solution (Na_3_PO_4_)

This experiment was performed using specimens A4-2 and A4-3. The specimens were immersed at room temperature for 24 h in Na_3_PO_4_ at molar concentrations of 0.25 M (A4-2-1 and A4-3-1), 0.5 M (A4-2-2 and A4-3-2), and 1.0 M (A4-2-3 and A4-3-3). The specimens then underwent three cycles of ultrasonic cleaning using purified water for 30 min, followed by oven drying for 24 h.

The formula for the Na_3_PO_4_ chemical reaction during immersion was as follows:CaCO_3_↓ + Na_3_PO_4_ → Ca_3_(PO_4_)_2_↓ + Na_2_CO_3_(6)

#### 4.2.3. Sintering Treatment

Sintering was performed on three groups of specimens (Z, A4-2-1, and A4-3-1) in a high-temperature furnace (DF-404, Dengyng Instruments Co., Ltd., New Taipei City, Taiwan) at a heating rate of 10 °C/min. In this study, sintering was performed for 2 h at the following temperatures: zirconia (1000 °C) and phosphate (1100 °C).

### 4.3. Material Characterization

#### 4.3.1. Characterization of Surface Morphology

The surface structures and element distributions in the specimens were characterized using a scanning electron microscope (SEM) (Hitachi SU 3500, Kyoto, Japan). Note that a thin layer of gold film (50 to 200 Å in thickness) was sputtered onto the material surface to facilitate characterization. Our primary focus was on surface holes caused by erosion during alkali treatment.

#### 4.3.2. Energy-Dispersive Spectrometry

We employed an SEM (5 kV) equipped using an energy-dispersive X-ray spectrometer (EDS; EX-250, HORIBA, Kyoto, Japan), with a focus on Zr, O, Na, Ca, and P at the surface. It operates at 5 kV and 60 Pa, with a metal coating.

#### 4.3.3. X-ray Photoelectron Spectroscopy

Chemical analysis was performed using an electron spectrometer (PHI 5000 VersaProbe, ULVAC-PHI) with an X-ray beam of <10 mm and an energy resolution of <0.50 eV (Ag3d5/2). C60 sputtering was used for surface analysis. Note that unlike conventional sputtering under Ar, C60 can perform chemical state correlation analysis without destroying the chemical structure of organic materials.

#### 4.3.4. Fourier-Transform Infrared Spectroscopy

A Nicolet iS5 FTIR (Thermo Scientific; Waltham, MA, USA) infrared light source was directed at a fixed mirror and a moving mirror to be combined into a single infrared light. Fourier transform infrared spectroscopy – attenuated total reflectance (FTIR–ATR) provides information related to the presence or absence of specific functional groups, as well as the chemical structure of polymer materials. Since zirconia tablets are opaque, we use semi-attenuated total reflection (ATR) for measurement.

#### 4.3.5. White Light Interferometry

White light interferometry was used to assess the surface roughness at the nanoscale level.

#### 4.3.6. Contact Angle Analyzer

A contact angle analyzer (Digidrop Goniometer; GBX, France) was used to determine the hydrophilicity of the materials. A 35 mm camera was used to capture images of water drops (4 mL) from which to derive the contact angle.

#### 4.3.7. X-ray Diffraction

X-ray diffraction (XRD, Panalytical X’Pert3 PRO, Malvern Panalytical Co. Ltd., Almelo, The Netherlands) was used to characterize crystalline structures in the specimens. We use an Ultima IV-Rigaku with a temperature chamber operating at 40 kV, 30 mA, with a CuKα radiation wavelength of λ = 1.5406 Å. The X-ray diffraction patterns were investigated from 20 to 80° on a 2θ scale with a step size of 0.02.

## 5. Conclusions

This study demonstrated that the sintering of zirconia can eliminate the occurrence of m-ZrO_2_, and that alkaline treatment can be used to obtain a superhydrophylic surface with highly uniform surface erosion to improve biocompatibility. The sequential immersion of zirconia specimens, respectively, in NaOH, Ca(OH)_2_, and Na_3_PO_4_ was shown to produce a uniform surface coating of OCP and B-type HAP, which are ideally suited to osseointegration.

## Figures and Tables

**Figure 1 ijms-23-06675-f001:**
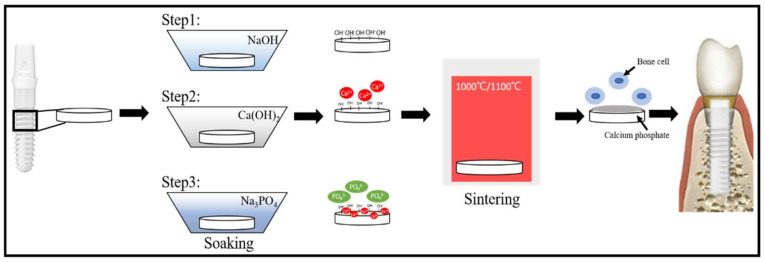
Graphic abstract of this study.

**Figure 2 ijms-23-06675-f002:**
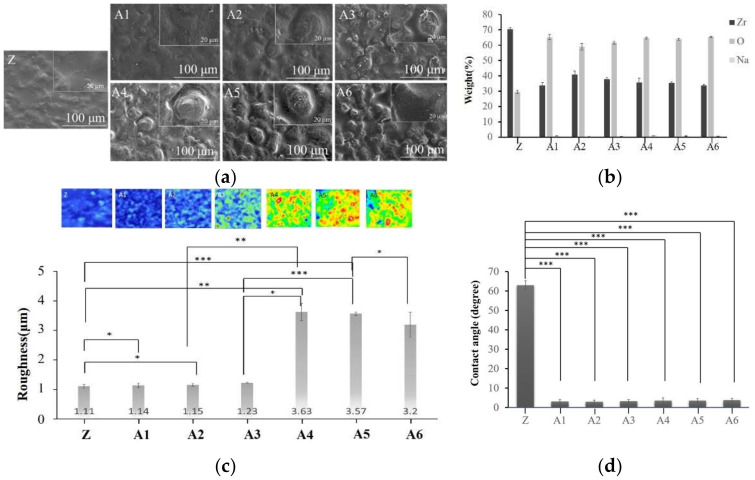
(**a**) The surface morphology of alkaline treatment samples was observed via SEM. (**b**) The elements of alkaline treatment samples were analyzed via EDS. (**c**) Surface roughness was analyzed via white light interferometry. (**d**) Comparison of the surface hydrophilicity after soaking in sodium hydroxide. (* *p* < 0.05; ** *p* ≤ 0.01; *** *p* < 0.001).

**Figure 3 ijms-23-06675-f003:**
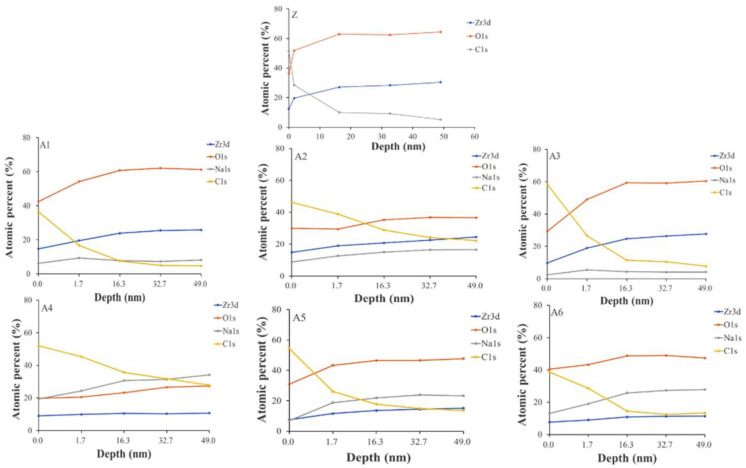
Elements of surface treatment were obtained via XPS.

**Figure 4 ijms-23-06675-f004:**
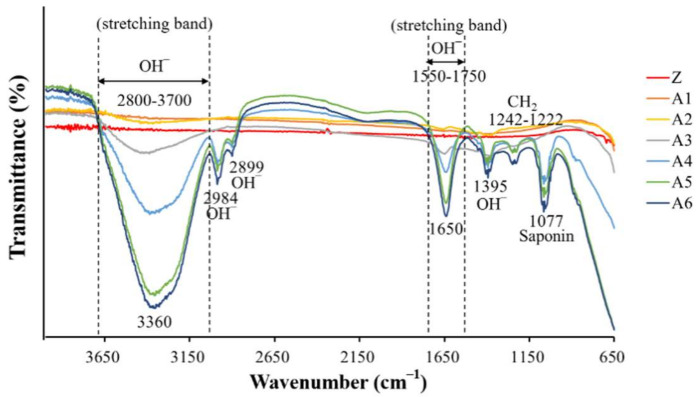
Fourier-transform infrared spectroscopy (FTIR). Functional groups of alkaline treatment samples were analyzed via FTIR.

**Figure 5 ijms-23-06675-f005:**
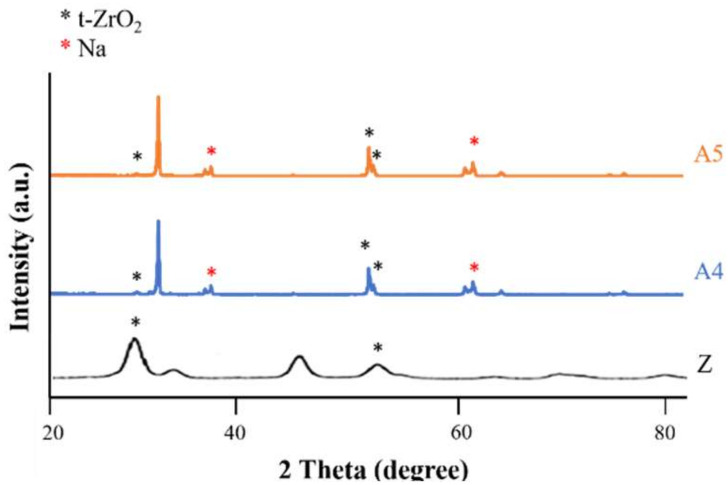
Use of XRD analysis for comparing untreated zirconia (Z) and alkaline treatments with different concentrations of A4 and A5.

**Figure 6 ijms-23-06675-f006:**
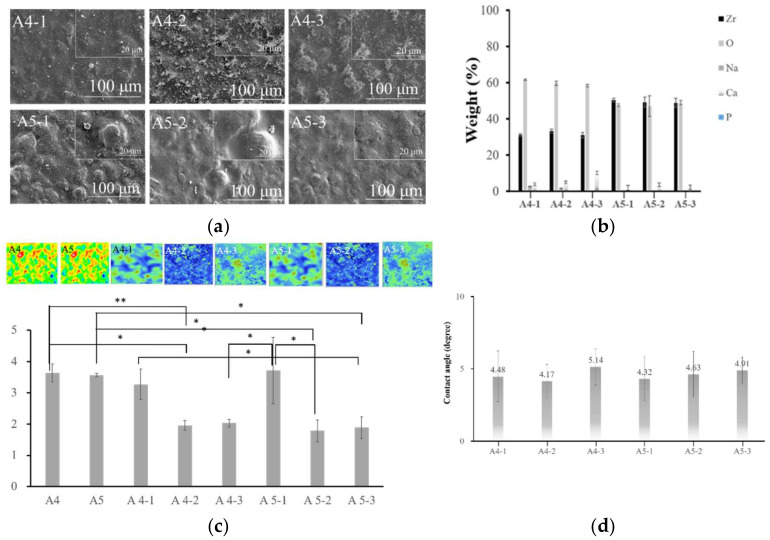
(**a**) Surface morphology of soaking in calcium hydride of A4 and A5. (**b**) Samples soaked in different concentrations of calcium hydroxide material surfaces at A4 and A5 substrates were analyzed via EDS. (**c**) Surface roughness analysis. Comparison of the surface roughnesses of A4 and A5 as the substrate after calcium hydroxide alkaline treatment (* *p* < 0.05; ** *p* < 0.01). (**d**) No significant difference. Comparison of the surface hydrophilicity after soaking in calcium hydroxide.

**Figure 7 ijms-23-06675-f007:**
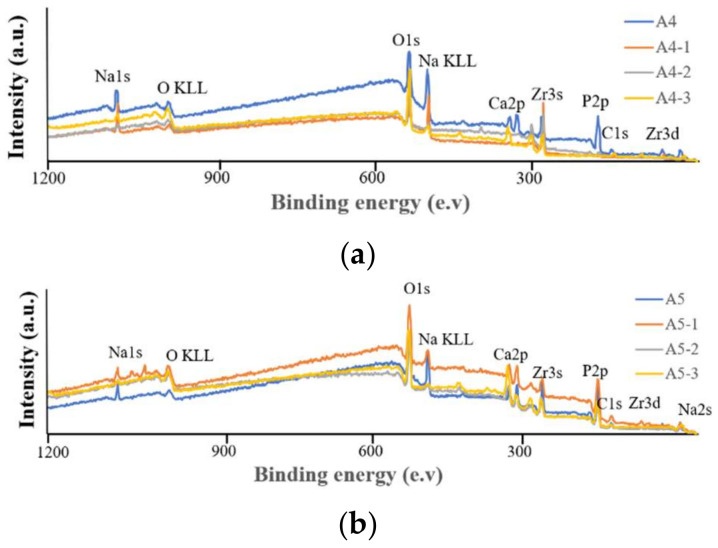
(**a**) XPS analysis of calcium hydroxide soaking, with A4 as the base. (**b**) XPS analysis of calcium hydroxide soaking, with A5 as the base.

**Figure 8 ijms-23-06675-f008:**
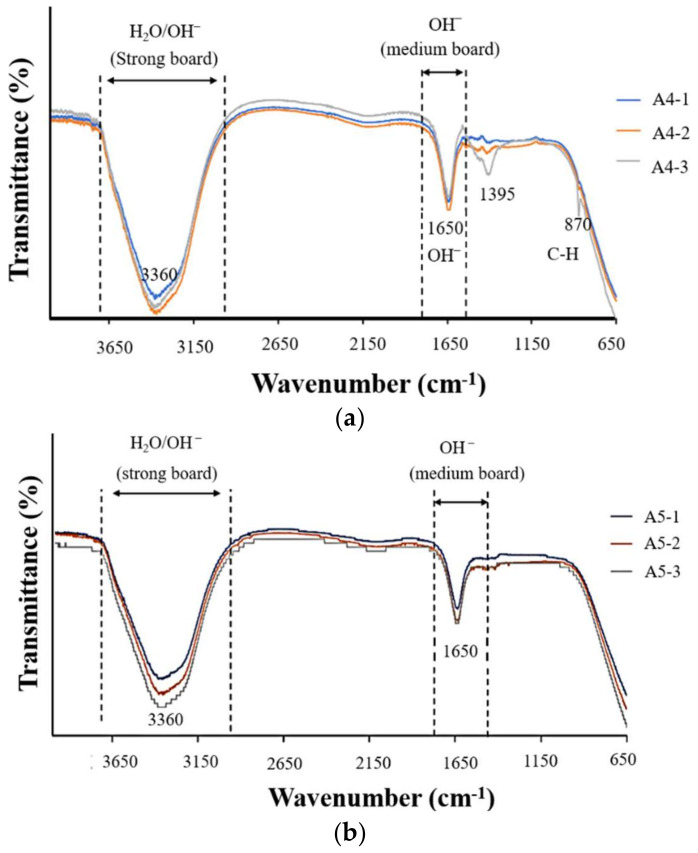
(**a**,**b**) FTIR was used to analyze the functional groups contained in the surface of calcium hydroxide-soaked materials, based on A4 (4-1, 4-2, 4-3) and A5 (5-1, 5-2, 5-3).

**Figure 9 ijms-23-06675-f009:**
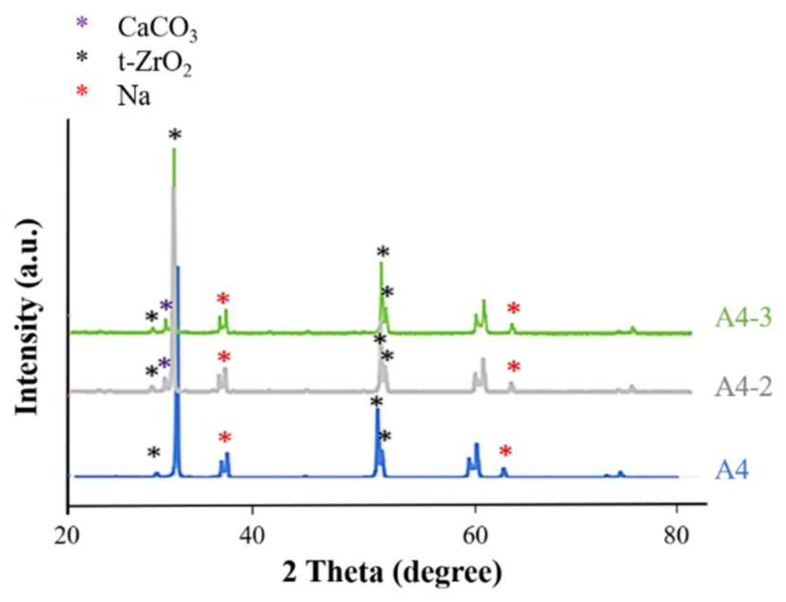
Alkali-treated zirconia (A4) was compared to A4-2 and A4-3 soaked with different concentrations of calcium hydroxide, using XRD analysis.

**Figure 10 ijms-23-06675-f010:**
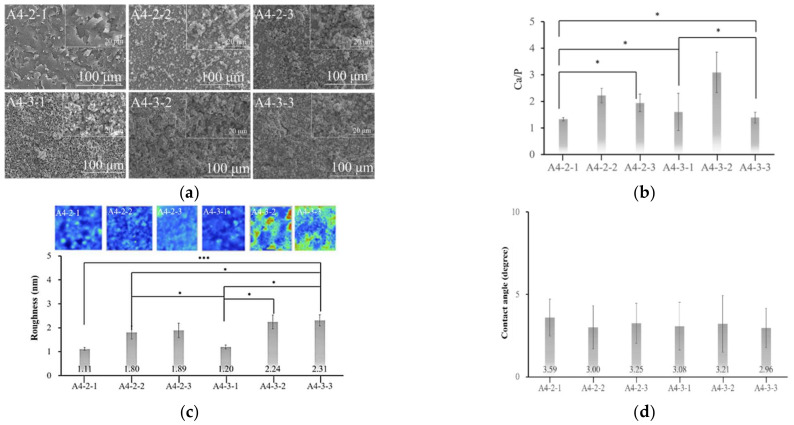
(**a**) The surface morphology of calcium phosphate was observed via SEM. (**b**) The calcium–phosphorus ratio (Ca/P) contained in the material’s surface can be measured separately via EDS, and the type of calcium phosphate can be inferred (* *p* < 0.05). (**c**) Surface roughness analysis. Comparing a different group of calcium phosphate (* *p* < 0.05; *** *p* < 0.001). (**d**) No significant difference. Comparison of the surface hydrophilicity after soaking in sodium phosphate.

**Figure 11 ijms-23-06675-f011:**
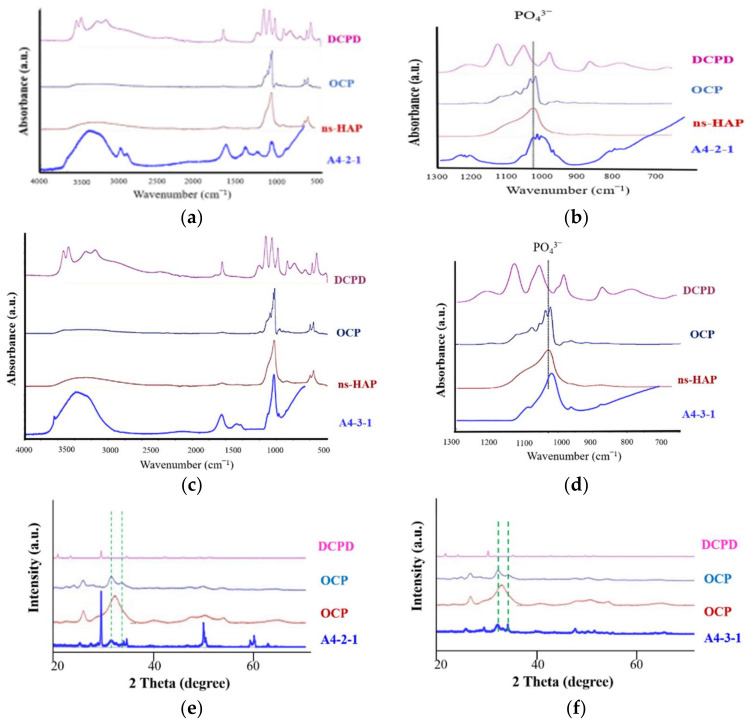
(**a**,**b**) A4-2-1 is compared to the crest of other calcium phosphate species via FTIR. (**b**) Detail of the 600–1300 degree interval in (**a**). (**c**,**d**) A4-3-1 compared with the crests of other calcium phosphate species via FTIR. (**d**) Detail of the 600–1300 degree interval in (**c**). (**e**) A4-2-1 is compared to the crest of other calcium phosphate species via XRD. (**f**) A4-3-1 is compared to the crest of other calcium phosphate species via XRD.

**Figure 12 ijms-23-06675-f012:**
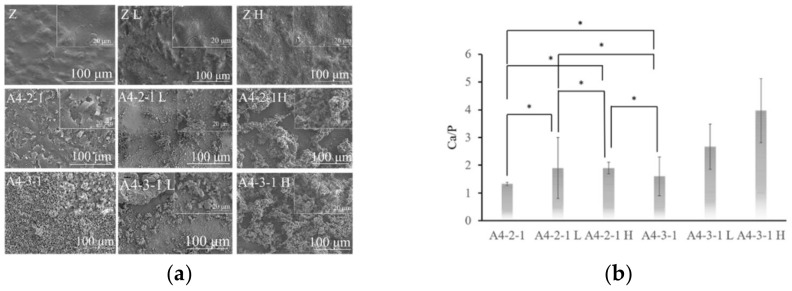
(**a**)The surface morphology of sintering samples was observed via SEM. (**b**) It was found by EDS that the calcium phosphorus compound was contained on the surface of the sintered material. The calcium-phosphorus ratio (Ca/P) shows the composition of calcium-phosphorus compounds (* *p* < 0.05).

**Figure 13 ijms-23-06675-f013:**
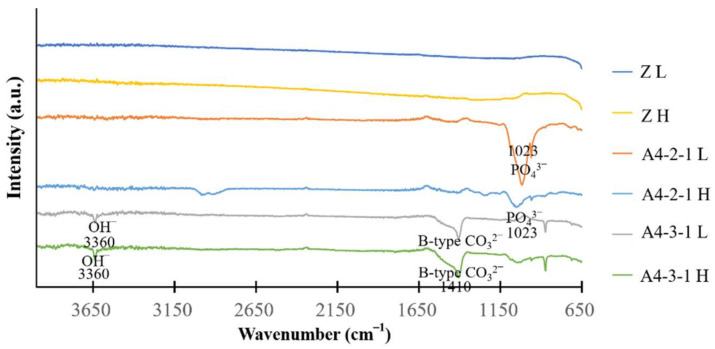
Fourier-transform infrared spectroscopy (FTIR). Functional groups of sintering samples were analyzed via FTIR.

**Figure 14 ijms-23-06675-f014:**
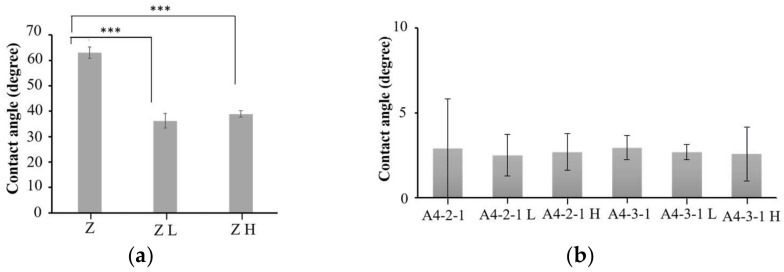
No significant difference. (**a**) Comparison of the surface hydrophilicity of untreated zirconia materials (*** *p* < 0.001). (**b**) Comparison of the hydrophilicity before and after A4-2-1 and A4-3-1 sintering.

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
