# Peer review of "Preparation of Calcium Phosphate Compounds on Zirconia Surfaces for Dental Implant Applications"

_ijms, 2022, doi:10.3390/ijms23126675_

Round 1
Reviewer 1 Report
Dear authors,
The manuscript is well written, and the study demonstrates the room-temperature fabrication of dental implants with super hydrophilic surfaces to enhance biocompatibility. The findings are interesting and the device could find application for dental implant.
It is recommended to accept the manuscript in its present form.
Thanking you,
Author Response
Dear Reviewer
Based on reviewer 1's comment, confirmation has been made.
No need to upload attachments.
Thanks for the detailed and quite helpful comment
We will work harder on research
Best Wishes
Reviewer 2 Report
The authors of this work presented the room-temperature fabrication of dental implants with super hydrophilic surfaces to improve biocompatibility. Zirconia, as the material being close to the color of teeth, is this material but its biological inertness hinders bonding with bone tissue. In avoidance of this performance, an alkaline treatment and coatings of calcium phosphate are often used to enhance bone regeneration close to the dental implant. In this paper, the authors used a surface treatment to overcome these questions. They applied the sintering of calcium phosphate to obtain a highly conducive effect and ease the bone integration.
This study demonstrated the room-temperature fabrication of dental implants with super hydrophilic surfaces to strengthen biocompatibility. There are the following quests to improve the paper:
(1) Fig. 7 and others, instead of A.U. on the vertical axis à use „a.u.”, as by convention
(2) Roughness (nm) à what roughness parameter do you mean? Is it Ra? You should indicate it!
(3) Line 77, >1300 degrees<, do you mean „deg. C”?
(4) Fig. 13, on the vertical axis, make space in „Contact angle (degree)”
(5) Separate your Mole, „2.5 M”, 5.0 M”, etc.
(6) Line 137 à see the PDF Enclosure
(7) Use MINUS instead of >semi-minus< in numerous places
(8) Line 413 à better „4 mL”
References à see the PDF Enclosure
Herewith please find the PDF Enclosure with the most important points highlighted in color – for corrections.

Author Response
Dear Reviewer
I have revised all the corrections you suggested.
All errors in our submitted article have been corrected based on the PDF you provided. The revised file is attached.
Thanks for the detailed and quite helpful comment. We will work harder on research.
Best Wishes
Response to Reviewer 2 Comments
Point 1: Fig. 7 and others, instead of A.U. on the vertical axis à use „a.u.”, as by convention
Response 1: All units in the figure have been modified (as attached). (in “Track Changes”)
Point 2: Roughness (nm) à what roughness parameter do you mean? Is it Ra? You should indicate it!
Response 2: All units in the figure have been modified (as attached). (in “Track Changes”)
Point 3: Line 77, >1300 degrees<, do you mean „deg. C”?
Response 3: All units in the figure have been modified (as attached). (in “Track Changes”)
Point 4: Fig. 13, on the vertical axis, make space in „Contact angle (degree)”
Response 4: All units in the figure have been modified (as attached). (in “Track Changes”)
Point 5: Separate your Mole, „2.5 M”, 5.0 M”, etc.
Response 5: All units in the figure have been modified (as attached). (in “Track Changes”)
Point 6: Line 137 à see the PDF Enclosure
Response 6: All units in the figure have been modified (as attached). (in “Track Changes”)
Point 7: Use MINUS instead of >semi-minus< in numerous places
Response 7: All units in the figure have been modified (as attached). (in “Track Changes”)
Point 8: Line 413 à better „4 mL”
Response 8: All units in the figure have been modified (as attached). (in “Track Changes”)
